# Factors Influencing SARS-CoV-2 Infection Control Practices of Nurses Caring for COVID-19 Patients in South Korea: Based on Health Belief Model

**DOI:** 10.3390/ijerph20043223

**Published:** 2023-02-12

**Authors:** Dain Jeong, Young Eun

**Affiliations:** 1College of Nursing, Gyeongsang National University, Jinju 52727, Republic of Korea; 2College of Nursing, Institute of Health Sciences, Gyeongsang National University, Jinju 52727, Republic of Korea

**Keywords:** COVID-19, health belief model, infection control, nurses, practice

## Abstract

This study aimed to verify the level of COVID-19 infection control practices and the factors affecting the COVID-19 infection control practices of Korean nurses based on the health belief model. The participants were 143 nurses experienced in caring for COVID-19 patients in South Korea. Questionnaires were used to measure health beliefs, confidence in practice, knowledge of COVID-19, infection protection environment, and COVID-19 infection control practices. Data were analyzed by performing descriptive statistics, an independent t-test, one-way analysis of variance, the Mann–Whitney test and multiple regression analysis. The mean score for infection control practices related to COVID-19 was 4.76 on a 5-point scale where a higher score indicates superior infection control performance. Multiple regression analysis revealed that the factors that influenced COVID-19 infection control practices were gender, marital status, perceived susceptibility, and confidence in practice related to COVID-19. With COVID-19 approaching endemic and to prevent infectious diseases, it is necessary to emphasize perceived sensitivity by providing accurate information on the risk of infection rather than simply inducing infection control to be divided into individual activities. In addition, nurses’ infection control practices should be implemented with confidence with the nurses themselves feeling the need for infection control and not being forced by the social atmosphere or the hospital.

## 1. Introduction

The coronavirus disease 2019 (COVID-19), a respiratory disease caused by a novel coronavirus, was declared a pandemic, the highest level of an epidemic warning, in March 2020 due to its rapid infection rate and social crisis [1]. To prevent further public harm, the Korea Disease Control and Prevention Agency (KDCA) designated a hospital for infectious diseases and divided the roles of public hospitals to isolating and treating patients with COVID-19 in negative pressure isolation rooms [2]. As confirmed patients flocked to these hospitals, the proportion of duties and roles of nurses who directly cared for the patients increased as they responded to the infectious disease outbreak [3]. COVID-19 is spread from person to person through droplets, contact and fomites. Therefore, hand hygiene, standard precautions to protect patients and medical staff, and appropriate use of personal protective equipment (PPE) are important to prevent respiratory infections [4].

On account of the nature of their roles, nurses often have direct contact with patients and frequently perform invasive medical procedures [5]. Therefore, their infection control practices play an important role in preventing infections [6]. In addition, the stress of nursing and the difficulties encountered by nurses are on the rise owing to having to adapt to new guidelines such as management of new patients with infectious disease, procedures for wearing protective equipment, infection control practices, and patient needs [7]. Infection control practices refer to activities to prevent the spread of infection by assessing infection-related risk factors and implementing preventive interventions in the process of nursing patients [8]. In Korea, the level of COVID-19 infection control practices was 82–96 points out of 100 points [9,10,11], representing a wide range. To control COVID-19 infections, more stringent infection control by nurses is important and required [5].

The health belief model (HBM) is used to explain health-related behaviors. It is also employed as a guiding framework for interventions that influence a specific behavior [12]. The HBM is identified as a factor that significantly influenced nurses’ infection control performance [9,10,13,14,15,16]. Research on nurses in an intensive care unit [13] and in general hospitals [16] found that a higher perceived benefit was associated with higher performance on multidrug-resistant organism (MDRO) infection control practices. In addition, a previous study on nurses in the emergency room [9] found that perceived severity and barriers were factors that affected their performance regarding COVID-19 infection control. To induce positive changes in an individual’s health, it is important to detect the risks and barriers of new infectious diseases [17] and confirm their effect on infection control practices.

In many investigations that used the HBM, self-efficacy was added as an important factor to predict an individual’s preventive behavior, which increased the HBM’s expansion and explanatory power. High self-efficacy can affect overall behavior such as the degree of effort to achieve a goal, the will to avoid risky behavior, and ways to overcome barriers [18]. Studies related to MDROs [19] investigated confidence in practicing infection control behaviors. It refers to one’s confidence in the ability to perform and execute infection control practices according to set procedures [19]. A previous study demonstrated that confidence in practicing infection control increased the MDRO practice rate in intensive care unit nurses [19]. In addition, new nurses’ high confidence levels enhanced their clinical performance [20]. Thus, confidence in practice is an important factor in improving nurses’ performance of COVID-19 infection control. In addition to health beliefs, knowledge of COVID-19 is essential for safe and high-quality nursing practice; incorrect knowledge affects performance and leads to increased transmission of the disease [21]. In the case of COVID-19 infection control, better knowledge regarding infectious diseases is related to higher levels of infection control practices [9].

Moreover, an infection prevention environment with material support at the hospital’s organizational level is important [22]. An infection prevention environment requires the provision of facilities, equipment, and administrative support related to prevention of exposure for both medical personnel and patients [23]. Infection control practices were higher when a safe work environment, appropriate supplies, and infection control guidelines were provided [24]. However, the infection prevention environment and implementation of infection control practices are also affected by COVID-19 [9]. Nevertheless, limited studies have reported on COVID-19 infection control practices based on the HBM while caring for COVID-19 patients. Therefore, based on the HBM, the purpose of this study were: (1) to examine the level of COVID-19 infection control practices, HBM variables, confidence in practice, knowledge of COVID-19, and infection prevention environment; (2) to examine the differences in infection control practices according to the general characteristics; and (3) to verify the factors affecting COVID-19 infection control practices of nurses in Korea.

## 2. Methods

### 2.1. Study Design

This study used a cross-sectional research design to investigate factors that affect clinical nurses’ COVID-19 infection control practices based on the HBM. The conceptual framework of this study is presented in Figure 1.

### 2.2. Participants

The participants were nurses who had cared for patients with COVID-19 in the negative pressure isolation rooms (NPIR). The inclusion criteria were nurses who had cared for COVID-19 patients since January 2020 in the NPIR and understood the purpose of the study. The sample size was calculated using the G*Power program 3.1.9.4, with an effect size of 0.15, a significance level of 0.05, a power of 90%, and six predictors. Considering a dropout rate of 20%, 150 nurses experienced in caring for COVID-19 patients were recruited. Data were collected from 15 November to 5 December 2021. Prior to data collection, the researcher visited the unit, explained the purpose of the study to the head nurse, and distributed the consent form, the questionnaire, and the reply envelope to the nurses. After the participants filled out the consent form, they completed the questionnaire and placed it in a sealed envelope. These envelopes were returned to the designated return box and collected by the researcher directly. The questionnaires were distributed to 150 nurses. After questionnaires with incomplete or missing data were excluded, the total number of participants was 143.

### 2.3. Questionnaires and Measurements

The questionnaire consisted of the following sections: (1) general characteristics of the participants, (2) COVID-19 infection control practices, (3) HBM variables, (4) confidence in practice, (5) knowledge of COVID-19, and (6) infection prevention environment. Overall, it comprised 97 questions and took approximately 20 min to complete.

The general characteristics were gender, age, marital status, education level, position, clinical experience, experience of education on COVID-19, previous experience of caring for patients during an epidemic, average work time in the COVID-19 NPIR, and the average number of COVID-19 patients cared for during duty. 

#### 2.3.1. COVID-19 Infection Control Practices

COVID-19 infection control practices were measured using the nurses’ infection control practices for new infectious diseases [11]. Items were modified and revised to respond to COVID-19 infection control practices for our analyses. To verify the validity of the items, the researcher requested the head nurse of the infection control room and COVID-19 unit, an infection control nurse, an intensive care unit nurse with more than 10 years of experience, and a professor from the nursing department to review the items. The initial measure consisted of 27 preliminary items. However, three items were deleted due to a content validity index (CVI) of ≤0.80. In addition, they did not meet the COVID-19 guidelines at the time. Therefore, the measure consisted of 24 items rated on a 5-point Likert scale (1 = never to 5 = always). A higher score indicated superior infection control performance. Yi and Cha [11] reported a Cronbach’s α of 0.90, and in this study, the Cronbach’s α was 0.92.

#### 2.3.2. Health Belief Variables

Health belief variables were measured using the health belief of management related to MDRO [13]. Items were modified for this study to use only COVID-19 terminology. The measure consisted of 19 items (perceived susceptibility, five items; perceived severity, four items; perceived benefits, three items; perceived barriers, four items; and cues to action, three items) rated on a 5-point Likert scale (1 = strongly disagree to 5 = strongly agree). A higher score indicated stronger belief in health. Kim and Cha [13] reported a Cronbach’s α of 0.79, and in this study, the Cronbach’s α was 0.78.

#### 2.3.3. Confidence in Practice

Confidence in practice was measured using the confidence in practice related to MDRO [19]. Items were modified for this study to use only COVID-19 terminology. There were 11 items rated on a 10-point Likert scale, and higher scores indicated higher confidence in practice. Choi [19] reported a Cronbach’s α of 0.97, and in this study, the Cronbach’s α was 0.92.

#### 2.3.4. Knowledge of COVID-19

Knowledge of COVID-19 was measured based on the COVID-19 situation as of November 2021 with reference to KDCA’s data [2]. To verify the content validity of the measure, the researcher asked the head nurse of the infection control room and COVID-19 unit, an infection control nurse, an intensive care unit nurse with more than 10 years of experience, and a professor from the nursing department to review the items. The initial measure consisted of 25 preliminary items. However, five items were deleted due to a CVI of ≤0.80. In addition, they did not fit well with the COVID-19 guidelines at the time. Therefore, the measure consisted of 20 items and 7 subdomains: clinical symptoms, chain of transmission, classification criteria, specimen management, PPE, standard precautions, and environment. The items were rated as 0 points (incorrect and unknown) or 1 point (correct). A higher score indicated better knowledge. Reliability of the measure was KR_20_ = 0.72.

#### 2.3.5. Infection Prevention Environment

The infection prevention environment was measured using the radiation protection environment [23]. Items were modified for our study to use only COVID-19 terminology. The measure consisted of 13 items rated on a 5-point Likert scale (1 = very poor to 5 = excellent). A higher score indicated a better infection prevention environment. Han and Moon [23] reported a Cronbach’s α of 0.79, and in this study, the Cronbach’s α was 0.92.

### 2.4. Data Analysis

Data analysis was conducted using SPSS WIN version 25.0. First, participant characteristics and levels of variables were analyzed using descriptive statistics. The normality test was verified by the Shapiro–Wilk test, skewness, and kurtosis. Second, the differences in infection control practices according to general characteristics were analyzed using the t-test and one-way analysis of variance. A Mann–Whitney test was used if normality was not tested. Finally, a multiple regression analysis was performed to identify the factors that affected nurses’ COVID-19 infection control practices.

### 2.5. Ethical Considerations

This study was approved by the Institutional Review Board (IRB No: 2021-09-021-002). Participation in the study was voluntary, anonymity was guaranteed, and the participants could withdraw from the study at any time. Furthermore, the participants were informed that their information would remain confidential and would be used only for research purposes. The purpose of the study was explained, and written informed consent was obtained. 

## 3. Results

### 3.1. Participant Characteristics

Of the 143 nurses in this study, 137 (95.8%) were female. The number of 25–29-year-olds was the highest, at 89 nurses (62.2%). Regarding marital status, 121 (84.6%) were single. Regarding education level, 89 nurses (62.2%) had a bachelor’s degree. The majority, 138 (96.5%), were staff nurses. Average clinical experience was 5.32 years, and 43 nurses (30.1%) had 5–10 years of experience. A total of 131 (91.6%) nurses had experience in education related to COVID-19, with continuing education being the most common (90.1%). A total of 117 (81.8%) nurses had no previous experience of caring for patients in epidemic nursing. During one duty cycle, the average working hours in the COVID-19 NPIR was 4.40 h, and the average number of COVID-19 patients cared for during duty was 4.48 (Table 1).

### 3.2. Level of Variables

The level of COVID-19 infection control practices was 4.76 ± 0.36 (5 points). The level of health belief was 3.85 ± 0.38 (5 points), and the subdomain levels of perceived susceptibility, severity, benefits, barriers, and cues to action were 3.79 ± 0.49, 3.94 ± 0.55, 4.20 ± 0.65, 3.90 ± 0.66, and 3.40 ± 0.76, respectively. The average level of confidence in practice was 8.62 ± 1.29 (10 points). In addition, the level of knowledge of COVID-19 was 17.48 ± 2.07 (20 points), and the level of infection prevention environment was 4.14 ± 0.61 (5 points) (Table 2).

### 3.3. Differences in Infection Control Practices According to General Characteristics

Differences in COVID-19 infection control practices according to general characteristics were gender (Z = −3.47, *p* = 0.001) and marital status (t = −3.74, *p* < 0.001). Specifically, practices were higher in female compared to male and in married nurses compared to unmarried nurses (Table 3).

### 3.4. Factors Affecting COVID-19 Infection Control Practices

To verify the factors affecting COVID-19 infection control practices, a stepwise multiple regression analysis was performed. Among the participants’ general characteristics, gender and marital status, which showed a significant difference, were treated as dummy variables. Perceived susceptibility, severity, benefits, knowledge of COVID-19, infection prevention environment, and confidence in practice, which showed significant correlations, were included in the multiple regression analysis.

The Durbin–Watson statistic was 1.91, close to 2, thus indicating that there is no first-order autocorrelation. The tolerance limit was 0.954–0.989 and the variance inflation factor was 1.014–1.048. Therefore, there was no problem of multicollinearity, and the assumption of no multicollinearity was satisfied.

The results of the multiple regression analysis indicated the factors that affected COVID-19 infection control practices were performance confidence (β = 0.37, *p* < 0.001), perceived susceptibility (β = 0.32, *p* < 0.001), gender (β = −0.27, *p* < 0.001), and marital status (β = −0.22, *p* = 0.001). In addition, the explanatory power of the model was 37.0% (F = 25.91, *p* < 0.001) (Table 4).

## 4. Discussion

This study analyzed factors that affect hospital nurses’ COVID-19 infection control practices, using the HBM as a conceptual foundation. Of the participants, 91.6% were experienced in education for the prevention and control of COVID-19, among which continuing education was the most common at 94.7%. Kim [9] reported that 89.5% of the nurses received education for the prevention and control of COVID-19, of which 42.9% received in-hospital education. Our results showed a higher proportion of education received compared to Kim’s findings. As Korea has suffered from both the H1N1 influenza and the Middle East respiratory syndrome, infection control in hospitals has been emphasized. In addition, the education of medical staff for future preparation has been essential. Therefore, education in all medical institutions has been emphasized due to the prolonged COVID-19 pandemic. Overseas, COVID-19-related education is being conducted through online learning platforms [25,26] provided through various networks.

In our study, health beliefs had the highest score for perceived benefits. This finding was similar to the effect of health beliefs on infection control practices among emergency room nurses [9], although the perceived benefits score in this study was lower. However, it was higher than the scores reported in a study of general hospital nurses [27]. It shows that the benefits of COVID-19 infection control practices for hospital nurses and for nurses caring for COVID-19 patients are generally perceived as such. Among the items of perceived benefits, “I believe that wearing PPE is helpful in preventing infection in nurses” had the highest score, which was the same in Kim [9]. Meanwhile, the score for perceived barriers to PPE was 4.23 points, which indicates that they felt uncomfortable. Wearing PPE causes physical discomfort and is a time-consuming process [27]. Nevertheless, it is necessary to emphasize the positive effects of PPE on personal health and safety and to spread awareness that COVID-19 can be prevented. It is also necessary to accommodate the extra time taken to properly wear PPE to prevent infection effectively.

The average confidence level in practice in our study was higher than the levels reported in a previous study [19], which measured the confidence in practice of MDRO infection control practices by nurses in general hospitals. Among the items, as in Choi et al. [19], hand hygiene scored the highest. These findings reveal that nurses recognize hand hygiene as the most basic and important infection prevention practice. Meanwhile, the score was low for “release from quarantine,” which highlights the need for immediate education on updated and recent guidelines.

The average score for knowledge of COVID-19 in this study was 17.48 points (20 points) (87.4%). Although the tools are different and it is difficult to make a direct comparison, previous studies [9,10,27], which measured knowledge of COVID-19 based on management guidelines of the KDCA, reported the average score for knowledge of COVID-19 was 84.9% [9], 70.2% [10], and 69% [21]. These findings indicate that nurses’ level of knowledge regarding COVID-19 is above average. In contrast, an overseas study on nurses’ knowledge of COVID-19 showed an average of 55–77.2% [28,29,30], thus indicating that the current study reported higher levels of knowledge in nurses. It may be because the participants in this study were nurses who cared for COVID-19 patients directly and were continuously exposed to information related to COVID-19. Among the items, the highest percentages were related to clinical characteristics, symptoms, and chain of transmission, similar to previous research [9,26]. However, post-management nursing had the lowest percentage. It was because, although there is a guideline for nursing after the death of a patient [2], the frequency of exposure to deaths was relatively low. Therefore, it is important to impart education on nursing practices after the death of a COVID-19 patient.

Quality of infection prevention environment in this study was 4.14 points (5 points). It was lower than 4.23 points (5 points) for emergency room nurses [9] and higher than 3.88 points for COVID-19 hospital nurses [31], which was measured using Han and Moon’s [23] tool. At the time of this study, the COVID-19 outbreak was prolonged, the provision of supplies and facilities for infection prevention were better prepared, and vaccinations had been implemented. According to our findings, the items on medical waste/laundry and PPE were high, which was similar to that of a previous study [31]. However, the number of health checkups of hospital staff was low. According to a study in China [32], healthcare workers pointed out improvements in the protection and health of medical staff. Therefore, it is necessary for hospital managers to continue paying attention to and securing finances to improve the infection prevention environment.

The COVID-19 infection control practices in this study were 4.76 points (5 points). Although we did not use the same tool, our results are higher than those of previous studies [9,10,11]. As mentioned earlier, the research institute was designated as an infectious disease center, and the number of confirmed cases were more than 5,000 at the time of the study in November 2021 [33]. Therefore, infection control practices were performed more frequently. The behaviors that showed the highest practice were medical waste disposal and putting on and taking off the PPE. However, the item that had the lowest score was for post-management practices. The most basic way to prevent infection was using the PPE. The KDCA also recommends that medical staff be trained on how to use and remove the PPE [2]. Medical waste has contagiousness of approximately 15%–20% and may pose a risk to exposed persons if managed improperly [34]. In addition, nurses are the most exposed to medical waste [35]. Secondary infection should be prevented through accurate and correct medical waste management based on COVID-19 guidelines. In addition, similar to the knowledge domain, post-management in infection control was also low. Hence, the importance of post-management should be emphasized to prevent secondary infections.

According to participant characteristics, we found that women performed infection control practices more than men, and married individuals performed the practices more than single individuals. Previous studies [9,11,28] showed no significant differences in gender and marital status. Therefore, it is necessary to conduct follow-up studies to evaluate general characteristics that affect infection control practices.

Based on the HBM, in this study, factors that affected COVID-19 infection control practices were determined using multiple regression analysis. The results indicated that confidence in practice, perceived susceptibility, gender and marital status contributed to COVID-19 infection control practices, in that order of potential influences. In addition, the explanatory power of the model was 37.0%. In a previous study [10], anxiety explained 6% of the variance in nurses’ COVID-19 infection control practices. In addition, Kim [9] demonstrated that knowledge, perceived severity, barriers, infection prevention environment, and monitoring of PPE contributed 32.7% of the variance in practices. In Kim et al. [36] with Korean adults, it was confirmed that gender and perceived susceptibility, benefits, and barriers influenced COVID-19 infection prevention behaviors. Compared to previous investigations, this study rationally explained infection control practices based on the HBM. Similar to the findings of previous study [36], we found perceived susceptibility as an influencing factor. The results of this study revealed that nurses were more likely to be infected with COVID-19 and were susceptive to themselves and healthy people. Nurses who take direct care of COVID-19 patients were afraid of being exposed to the infection or transmitting it to others and family members [37]. In addition, they had persistent anxiety regarding the infection. Owing to the prolonged COVID-19 pandemic, awareness of the infection has increased due to the media, rising cases, and through direct or indirect exposure to COVID-19 patients. 

Gender and marital status are factors affecting COVID-19 infection control practices. Female nurses are more aware of the risk of infection and are better at preventing infection. However, since only 4.2% of nurses in this study were male, caution should be taken in generalizing the findings. Furthermore, married nurses have higher infection control practices than unmarried nurses. This affirms that married nurses might be sensitive and more careful in regard to preventing transmitting the disease to their families [36]. 

In this study, knowledge of COVID-19 and perceived benefits were not identified as influencing factors for infection control practices. However, findings from previous research showed that knowledge [9,27] and perceived benefits [27,36] influenced practices and explained 17.1–35.0% of the variance. In addition to infection control practices, they reported that perceived benefits [13,16] was a factor that affected MDRO infection management, explaining 28.4–35.0% of the variance. These findings reveal that health beliefs are an important factor that contribute to infection control practices. It is difficult to directly compare the results due to a lack of previous research on confidence in practice. However, it could be interpreted as similar to previous study [38], where confidence in performing core basic nursing practices in nursing students was a factor that influenced clinical practice. As confidence in practice increases, learning effects are enhanced through increased motivation to be active and voluntarily engage in more active learning [39]. Hence, measures to increase confidence through education are required. In previous studies, confidence in nursing performance was gained through simulation [40]. Therefore, specific and repetitive education reflecting the COVID-19 guidelines should be implemented. In addition, education should emphasize the susceptibility of COVID-19 and the benefits of infection control practices using simulation.

This study has a few limitations to be considered when interpreting the findings. First, our study concerned nurses who cared for COVID-19 patients in a tertiary general hospital. Therefore, the results should be generalized with caution. Second, only perceived susceptibility and confidence in practice were found to be influencing factors, and we suggest follow-up studies on additional influencing factors that affect COVID-19 infection control practices.

## 5. Conclusions

This study is meaningful in verifying the factors that affect the infection control practice of COVID-19 based on the HBM during the COVID-19 pandemic. In addition, it is also valuable for nurses directly caring for COVID-19 patients in the NPIR. Based on HBM, it was found that nurses performed infection control based on their own beliefs. Therefore, it is necessary to emphasize perceived sensitivity by providing accurate information on the risk of infection rather than simply causing infection control to be divided into individual activities. In addition, nurses’ infection control practices should be implemented with confidence, with the nurses themselves feeling the need for infection control and not being forced by the social atmosphere or the hospital. To prevent COVID-19 from becoming endemic and causing new epidemics, nurses must continue to implement infection control practices.

## Figures and Tables

**Figure 1 ijerph-20-03223-f001:**
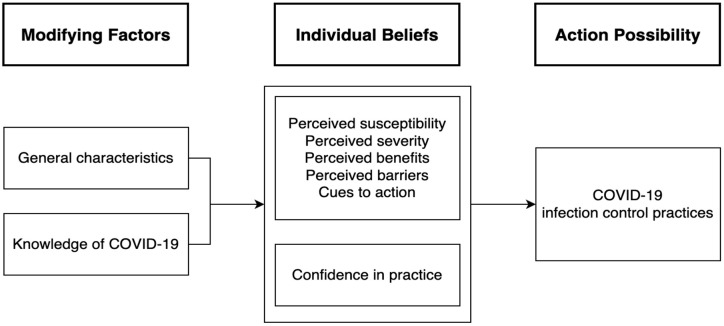
Conceptual framework of this study.

**Table 1 ijerph-20-03223-t001:** Participants’ characteristics (*n* = 143).

Characteristics	Categories	*n*	%	Mean ± SD
Gender	Male	6	4.2	
	Female	137	95.8	
Age (year)	≤24	19	13.3	
	≥25–<30	89	62.2	
	≥30–<35	19	13.3	
	≥35	16	11.2	
Marital status	Unmarried	121	84.6	
	Married	22	15.4	
Education level	Associate degree	41	28.7	
	Bachelor degree	89	62.2	
	Master degree	13	9.1	
Position	Staff nurse	138	96.5	
	Charge nurse	5	3.5	
Clinical experience (years)	<1	11	7.7	5.32 ± 4.43 (years)
	≥1–<3	31	21.7	
	≥3–<5	42	29.4	
	≥5–<10	43	30.1	
	≥10–<20	16	11.2	
Experience of education on COVID-19	Yes	131	91.6	
	No	12	8.4	
Form of education on COVID-19 (*n* = 131)	Continuing education	118	90.1	
	Department education	6	4.6	
	Other	7	5.3	
Previous experience of caring patients in epidemic	Yes	26	18.2	
	No	117	81.8	
Average work time with COVID-19 NPIR per duty (hours)				4.40 ± 1.00 (hours)
Average number of caring COVID-19 patients during duty				4.48 ± 1.96
Abbreviations: SD = standard deviation, NPIR = negative pressure isolation room				

**Table 2 ijerph-20-03223-t002:** Level of variables (*n* = 143).

Variables	Mean ± SD	Min	Max
COVID-19 infection control practices	4.76 ± 0.36	2.96	5.00
Health beliefs	3.85 ± 0.38	2.79	4.74
Perceived susceptibility	3.79 ± 0.49	2.00	5.00
Perceived severity	3.94 ± 0.55	2.50	5.00
Perceived benefits	4.20 ± 0.65	2.00	5.00
Perceived barriers	3.90 ± 0.66	1.75	5.00
Cues to action	3.40 ± 0.76	1.67	5.00
Confidence in practice	8.62 ± 1.29	4.09	10.00
Knowledge of COVID-19	17.48 ± 2.07	6.00	20.00
Infection prevention environment	4.14 ± 0.61	2.23	5.00
Abbreviation: SD = standard deviation			

**Table 3 ijerph-20-03223-t003:** Differences in infection control practices according to general characteristics (*n* = 143).

Characteristics	Categories	*n*	%	Mean ± SD	COVID-19 Infection Control Practices
					t/F (*p*)
Gender ^1^	Male	6	4.2	4.33 ± 0.25	−3.47 (0.001)
	Female	137	95.8	4.78 ± 0.35	
Age (year)	≤24	19	13.3	4.60 ± 0.42	1.74 (0.162)
	≥25–<30	89	62.2	4.79 ± 0.03	
	≥30–<35	19	13.3	4.75 ± 0.62	
	≥35	16	11.2	4.86 ± 0.23	
Marital status	Unmarried	121	84.6	4.74 ± 0.38	−3.74 (<0.001)
	Married	22	15.4	4.91 ± 0.13	
Education level	Associate degree	41	28.7	4.79 ± 0.22	1.50 (0.225)
	Bachelor degree	89	62.2	4.73 ± 0.42	
	Master degree	13	9.1	4.91 ± 0.18	
Position ^1^	Staff nurse	138	96.5	4.76 ± 0.37	−1.74 (0.081)
	Charge nurse	5	3.5	4.97 ± 0.46	
Clinical experience (years)	<1	11	7.7	4.71 ± 0.17	1.61 (0.176)
	≥1–<3	31	21.7	4.66 ± 0.48	
	≥3–<5	42	29.4	4.82 ± 0.20	
	≥5–<10	43	30.1	4.75 ± 0.44	
	≥10–<20	16	11.2	4.89 ± 0.20	
Experience of education on COVID-19	Yes	131	91.6	4.76 ± 0.37	−0.94 (0.925)
	No	12	8.4	4.77 ± 0.20	
Previous experience of caring patients in epidemic	Yes	26	18.2	4.73 ± 0.53	−0.51 (0.613)
	No	117	81.8	4.77 ± 0.32	

^1^ = Mann–Whitney test, Abbreviation: SD = standard deviation.

**Table 4 ijerph-20-03223-t004:** Factors affecting COVID-19 infection control practices (*n* = 143).

	B	SE	β	t	*p*	Collinearity	
						Tolerance	VIF
Constant	3.20	0.23		14.00	<0.001		
Confidence in practice	0.10	0.02	0.37	5.46	<0.001	0.958	1.044
Perceived susceptibility	0.23	0.05	0.32	4.66	<0.001	0.954	1.048
Gender (Ref: Female)	−0.48	0.12	−0.27	−4.00	<0.001	0.989	1.011
Marital status (Ref: Married)	−0.22	0.07	−0.22	−3.25	0.001	0.986	1.014
R^2^ = 0.388, Adjusted-R^2^ = 0.370, F = 21.86, *p* < 0.001, Durbin–Watson = 1.91							
Abbreviations: SE = standard error, VIF = variance inflation factor							

## Data Availability

The data presented in this study are available from the corresponding author upon reasonable request.

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
