# Peer review of "Factors Influencing SARS-CoV-2 Infection Control Practices of Nurses Caring for COVID-19 Patients in South Korea: Based on Health Belief Model"

_ijerph, 2023, doi:10.3390/ijerph20043223_

Round 1
Reviewer 1 Report
First, I want to congratulate the authors for submitting the manuscript.
The manuscript is a cross-sectional study, carried out in South Korea, which aims to investigate the factors that affect the control and practices of COVID-19 infection of clinical nurses based on the health belief model. In fact, the infection is not COVID-19 (since COVID-19 is the disease), but SARS-CoV-2, so I suggest changing the title to "Factors influencing SARS-Cov-2 infection control practices of nurses caring for COVID-19 patiens in South Korea: Based on Health Belif model".
References are current and relevant to the study.
Regarding the introduction:
Line 35: The statement "and close contact with infected individuals" can lead to confusion with contact transmission.
Line 44–47: The statement "In Korea, the level of COVID-19 infection control practices was 82–96 points out of 100 points, and strict infection control by nurses is important and required to control COVID-19 infections" must be framed , because you don't understand what the "level of infection control practices" is.
Regarding the methods:
Line 112: Is it possible to answer a 97-question questionnaire in 20 minutes?
Line 131–136: Why use a questionnaire that assesses the health belief of management related to multidrug-resistant organism, if COVID-19 is a disease caused by a virus?
Lines 162–167: Statistical options must be made explicit. Why were parametic tests used if, for example, in the gender variable there are only 6 males? Have assumptions been checked for using parametic tests? Were assumptions made for the regression analysis?
Regarding the results:
Line 207–209: in the presentation of results, it is not possible to verify the statement "Perceived susceptibility, severity, benefits, knowledge of COVID-19, infection prevention environment, and confidence in practice, which showed significant correlations". If these variables were included in the regression analysis, Table 4 does not show the results for "severity", "benefits", "knowledge of COVID-19", and "infection prevention environment".
Regarding the discussion:
If gender and marital status are factors affecting COVID-19 infection control practices, what are the implications for clinical practice?
Regarding the conclusion:
The answer to the objective of the study should be presented.
Author Response
Thank you for your comment concerning our manuscript. We sincerely appreciate the reviewer’s comments. We authors had meetings to discuss reviewer’s comments on our article. We revised our paper at our best on your comments. These comments were all valuable and very helpful for revising and improving our paper. All changes were marked in red in the manuscript.

Reviewer 2 Report
I inform that the work remains without a clear identification of the objective of the work, both in the abstract and at the end of the introduction, which is recommended for works of a scientific nature, which makes it impossible, for example, to assess whether the conclusion weaves enough robustness in the face of the goal. In addition, the statistical analysis is not explained in detail, just informing that it is descriptive statistics in the first art does not manage to present the readers with the format of the results, when you decide to use a standard descent mean, I ask: did you use a normality test To adopt this statistic? And the same happened for the use of the T test? Therefore, it is essential that these doubts are resolved so that the work is actually evaluated for qualification and possible publication.
Author Response
Response to Reviewer
Date: Jan 28, 2023
Ijerph-2112362
Thank you for your comment concerning our manuscript. We sincerely appreciate the reviewer’s comments. We authors had meetings to discuss reviewer’s comments on our article. We revised our paper at our best on your comments. These comments were all valuable and very helpful for revising and improving our paper. All changes were marked in red in the manuscript.

Round 2
Reviewer 1 Report
Thank you for your effort in responding to the comments made.